# A Case Report on a Human Bite Contact with a Rabid Honey Badger *Mellivora capensis* (Kromdraai Area, Cradle of Humankind, South Africa)

**DOI:** 10.3390/tropicalmed8040186

**Published:** 2023-03-24

**Authors:** Debrah Kgwana Mohale, Ernest Ngoepe, Munangatire Mparamoto, Lucille Blumberg, Claude Taurai Sabeta

**Affiliations:** 1World Organisation for Animal Health Rabies Reference Laboratory, Agricultural Research Council, Onderstepoort Veterinary Institute, Onderstepoort, Pretoria 0110, Gauteng, South Africa; 2Division for Outbreak Preparedness and Response and the Centre for Emerging, Zoonotic and Parasitic Diseases, National Institute of Communicable Diseases, 1 Modderfontein Road, Johannesburg 2192, Gauteng, South Africa; 3Department of Veterinary Tropical Diseases, Faculty of Veterinary Sciences, University of Pretoria, Onderstepoort, Pretoria 0110, Gauteng, South Africa

**Keywords:** rabies, lyssavirus, honey badger, wildlife, human

## Abstract

In South Africa, rabies cycles are sustained by both domestic and wildlife host species. Despite the fact that the majority of human rabies cases are associated with dog bite exposures, wildlife species can potentially transmit rabies virus (RABV) infection to humans. In July 2021, a honey badger (*Mellivora capensis*) from the Kromdraai area (Gauteng Province) bit a dog on a small farm. The following day the same honey badger attacked three adults in the area, with one of the victims requiring hospitalization for management of her injuries. The honey badger was subsequently shot and the carcass submitted to the Agricultural Research Council-Onderstepoort Veterinary Research (ARC-OVR) for RABV diagnosis. A positive rabies diagnosis was confirmed and phylogenetic analysis of the amplified glycoprotein gene of the rabies virus demonstrated the virus to be of dog origin.

## 1. Introduction

Rabies virus (RABV) is the causative agent of a neglected and encephalitic disease that results in an estimated 59,000 human deaths annually, primarily in resource-limited areas of Asia and Africa [1]. *Lyssavirus rabies*, a prototype species in the *Lyssavirus* genus (*Rhabdoviridae* family, order *Mononegavirales*), is associated with a public health hazard of the disease globally, accounting for at least 95% of the dog-mediated human rabies deaths [2]. The *Lyssavirus* genus consists of 17 viral species [3], and the members of this genus were delineated into three distinct phylogroups (I, II and III) based on biological and genetic characteristics [4,5]. Kotalahti bat lyssavirus (KBLV), a putative species identified in a Brandt’s bat (*Myotis brandtii*) in Finland, and Matlo bat lyssavirus [6] recovered from the brain of a Natal long-fingered bat (*Miniopterus natalensis*) in South Africa, are both awaiting formal recognition by the International Committee of the Taxonomy of Viruses. 

RABV has a single, non-segmented negative-strand RNA genome encoding information for five viral proteins, these are the nucleoprotein (N), phosphoprotein (P), matrix protein (M), glycoprotein (G) and large protein (L). Of these, G is a component of the viral envelope and facilitates viral attachment to the host cell surface, receptor binding and membrane fusion. In addition, G also influences viral pathogenicity and neurotropism, inducing the production of virus-neutralizing antibodies. Thus, the G protein is pivotal in determining RABV entry and adaptation to the host. 

In South Africa, rabies is maintained in both domestic (urban) and wildlife host species (sylvatic). Domestic dogs (*Canis familiaris*) are the reservoir and principal vector species of rabies, and through their interactions with wildlife species [(black-backed jackals (*Canis mesomelas*) and bat-eared foxes (*Otocyon megalotis*)] maintain and easily exchange the canid rabies variant [7,8,9]. Cross-species transmission events (CSTEs) are common and result in the exchange of canid RABVs between domestic and wildlife host species. The role and specific contributions of wildlife host species in CSTEs towards the public health hazard of the disease has not been fully elucidated. The dog rabies variant was originally introduced in the sub-region in the late 1940s (apparently from Angola); it is now well-established in wildlife host species, confirming the opportunistic nature of this lyssavirus variant [10]. Findings from a study performed by [11] demonstrated that 99% of the human rabies cases (*n* = 105) in South Africa were linked to dog exposures. In this study, we reconstructed the phylogenetic history using a partial region of the glycoprotein and the G-L intergenic region of domestic dogs and wildlife hosts (a honey badger and black-backed jackal species) and found that the RABVs from both the wildlife and domestic (dogs) belong to one distinct cluster. These findings not only underscore the importance of dog rabies in disease epizootiology but also has implications for its control in rabies-endemic areas of South Africa.

## 2. Materials and Methods

### 2.1. Specimens and Rabies Testing

There are many game farms in the Kromdraai area (Gauteng Province), and these are known to have been affected by jackal rabies with subsequent spill-over into domestic dogs and livestock in that area. On 25 July 2021, a honey badger was observed in the Kromdraai area and bit a dog on a farm. This area is situated in the Cradle of Humankind, a national heritage site within the ‘Cradle of Humankind’ (Figure 1). The following day, the honey badger in question attacked three women in the area, resulting in one of them being hospitalized for management of severe injuries. All victims received timeous post-exposure rabies prophylaxis with wound infiltration and human rabies immunoglobulin (HRIG) and rabies vaccine, but the hospitalized patient succumbed as a result of her injuries. The owner of the neighboring property shot the honey badger and the carcass was submitted to the Agricultural Research Council-Onderstepoort Veterinary Research (ARC-OVR) for rabies testing.

The carcass of the honey badger was submitted to the ARC-OVR on 27 July 2021 for rabies testing and the specimen was assigned a laboratory reference number (192/21) (Table 1 contains epidemiological information of the honey badger and the domestic and wildlife rabies viruses analyzed here). The appropriate brain parts were extracted for rabies testing using the gold standard direct fluorescent antibody test (DFA). A composite smear was prepared and, together with a positive and negative control, the smears were stained with a fluorescein isothiocyanate (FITC)-labelled polyclonal antibody preparation (Onderstepoort, Pretoria, South Africa) as described previously [12]. Subsequent to two PBS washes (in 0.1 M, pH 7.2–7.4) and counter-staining with Evans Blue (Sigma-Aldrich, St. Louis, MO, USA), the slides were examined under ultra-violet fluorescence (Zeiss, Berlin, Germany) by two experienced analysts. The results were recorded as positive (on a scale from +1 (lyssavirus antigen present in 25% of the field of the smear) to +4 (lyssavirus antigen present in 100% of the field) and as negative (−) in the absence of any fluorescing viral particles [13].

**Table 1 tropicalmed-08-00186-t001:** Epidemiological information of rabies viruses genetically characterized in this study.

Virus #	Laboratory Number	Species	Geographic Location	Accession Number	Reference
1	137/21	Black-backed jackal	South Africa, Gauteng Province	OP939356	This study
2	144/21	Black-backed jackal	South Africa, Gauteng Province	OP939357	This study
3	159/21	Black-backed jackal	South Africa, Gauteng Province	OP939358	This study
4	169/21	Black-backed jackal	South Africa, Gauteng Province	OP939359	This study
5	170/21	Black-backed jackal	South Africa, Gauteng Province	OP939360	This study
6	192/21	Honey badger	South Africa, Gauteng Province	OP939353	This study
7	217/21	Domestic dog	South Africa, Gauteng Province	OP939355	This study
8	242/21	Black-backed jackal	South Africa, Gauteng Province	OP939354	This study
9	115/11	Domestic dog	South Africa, Gauteng Province	JN227482	[12]
10	157/11	Domestic dog	South Africa, Gauteng Province	JN227483	[12]
11	288/11	Domestic dog	South Africa, Gauteng Province	JN227487	[12]
12	343/11	Domestic dog	South Africa, Gauteng Province	JQ756150	[12]
13	464/10	Domestic dog	South Africa, Gauteng Province	JF327493	[12]
14	503/10	Domestic dog	South Africa, Gauteng Province	JF327494	[12]
15	409/16	Black-backed jackal	South Africa, Gauteng Province	MW413411	[14]
16	428/16	Black-backed jackal	South Africa, Gauteng Province	MW413414	[14]
17	489/16	Bovine	South Africa, Gauteng Province	MW413403	[14]
18	527/16	Bovine	South Africa, Gauteng Province	MW413405	[14]
19	769/16	Black-backed jackal	South Africa, Gauteng Province	MK103257	[15]
20	784/16	Black-backed jackal	South Africa, Gauteng Province	MK103273	[15]
21	454/17	Black-backed jackal	South Africa, North-West Province	MT454646	[16]
22	460/17	Black-backed jackal	South Africa, North-West Province	MT454647	[16]
23	466/17	Black-backed jackal	South Africa, North-West Province	MT454648	[16]
24	474/17	Black-backed jackal	South Africa, North-West Province	MT454649	[16]
25	480/17	Black-backed jackal	South Africa, North-West Province	MT454651	[16]
26	14/377	Domestic dog	South Africa, Eastern Cape Province	MF197279	[17]
27	14/392	Domestic dog	South Africa, Eastern Cape Province	MF197280	[17]
28	14/430	Domestic dog	South Africa, Eastern Cape Province	MF197227	[17]
29	15/474	Domestic dog	South Africa, Eastern Cape Province	MF197283	[17]
30	16/318	Domestic dog	South Africa, Eastern Cape Province	MF197284	[17]
31	16/388	Domestic dog	South Africa, Eastern Cape Province	MF197285	[17]
32	10/299	Domestic dog	South Africa, KwaZulu Natal Province	KC660351	[18]
33	10/479	Domestic dog	South Africa, KwaZulu Natal Province	KC660348	[18]
34	88/08	Domestic dog	South Africa, Limpopo Province	GU808517	[19]
35	169/09	Domestic dog	South Africa, Limpopo Province	GU808515	[19]
36	460/09	Domestic dog	South Africa, Limpopo Province	GU808514	[19]
37	572/09	Domestic dog	South Africa, Limpopo Province	GU808513	[19]
38	1305/09	Domestic dog	South Africa, Limpopo Province	GU808512	[19]
39	201/12	Bovine	Lesotho, Maseru	MF197287	[17]
40	60/14	Bovine	Lesotho, Maseru	MF197297	[17]
41	195/14	Bovine	Lesotho, Berea	MF197299	[17]
42	17/15	Domestic dog	Lesotho, Maseru	MF197305	[17]
43	21/15	Bovine	Lesotho, Maseru	MF197303	[17]
44	24/15	Domestic dog	Lesotho, Maseru	MF197304	[17]
45	572/99	Domestic dog	Mozambique, Gaza Province	KM262039	[17]
46	558/05	Domestic dog	Mozambique, Inhambane Province	KM262043	[17]
47	482/12	Domestic dog	Mozambique, Maputo Province	KM262047	[17]
48	233/13	Domestic dog	Mozambique, Maputo Province	KM262049	[20]
49	804/06	Domestic dog	Mozambique, Manica	EU123934	[20]
50	187/14	Domestic dog	Zimbabwe, Harare	MF425794	[20]
51	200/14	Domestic dog	Zimbabwe, Harare	MF425796	[20]
52	355/14	Domestic dog	Zimbabwe, Harare	MF425800	[20]
53	239/15	Domestic dog	Zimbabwe, Harare	MF425805	[19]
54	406/15	Domestic dog	Zimbabwe, Harare	MF425808	[19]
55	464/15	Domestic dog	Zimbabwe, Harare	MF425811	[19]

**Figure 1 tropicalmed-08-00186-f001:**
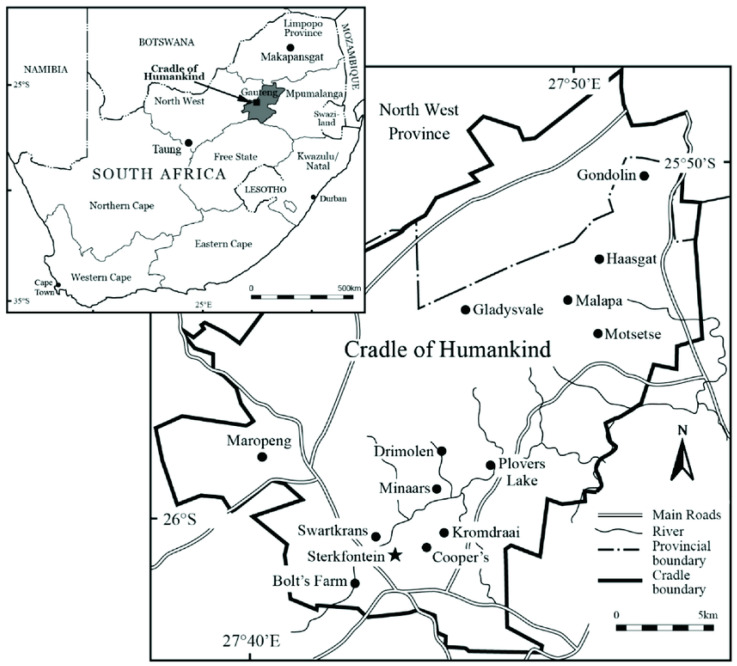
Map of South Africa showing the geographic origin of Kromdraai area [21]. Reprinted with permission from ref. [21], Copyright 2019, The South African Archaeological Society.

### 2.2. Antigenic Typing

Monoclonal antibody typing of the lyssavirus recovered from the honey badger (#192/21) was performed using a panel of 16 murine anti-lyssavirus nucleocapsid monoclonal antibodies (mAbs) and compared with other lyssaviruses from domestic and black-backed jackal species from the same geographical locality as the honey badger RABV (Table 2). The anti-N mAbs were generously donated by Dr Christine-Fehlner Gardiner (Head of the Centre of Expertise for Rabies, Canadian Food Inspection Agency, Ottawa, Canada). The panel of mAbs comprised 14 anti-rabies virus nucleoprotein mAbs, an anti-human adenovirus type-5 mAb (1C5) as a negative control and a positive control (38HF2). The panel was anti-N mAbs and could differentiate southern African lyssaviruses and variants [21]. The reactivity for each mAb was recorded either as positive (on a scale from 1 to 3) or negative (−) to generate the overall staining pattern for each lyssavirus [22].

### 2.3. Total RNA Extractions, Reverse Transcription PCR (RT-PCR), Nucleotide Sequencing and Phylogenetic Reconstruction

Total RNA was extracted from approximately 100 ng of brain-infected tissues of the honey badger and other local RABVs recovered from domestic (dogs) (*n* = 2) and black-backed jackal species (*n* = 4) (see Table 1) using TriReagent^R^ (Sigma Aldrich, St. Louis, MO, USA). The RNA was quantified using a nanodrop spectrophotometer.

The G (+) and L (−) primers were used in the RT–PCR reactions. The oligonucleotides, G (+) and L (−), respectively, 46655GAC TTG GGT CTC CCG AAC TGG GG34687 and 55205CAA AGG AGA GTT GAG ATT GTA GTC35543, were synthesized by Inqaba Biotechnical Industries (Pty) Ltd. (Pretoria, South Africa) at 100 nmol concentration. Prior to use, the oligonucleotides were reconstituted in 1X TE buffer [pH 8.0] to 100 pmoles/µL and then diluted 10-fold for use in cDNA synthesis and PCR assays [23]. The primers were used without any further purification. The annealing positions and polarity of the G (+) and L (−) primers were designated according to that of the Pasteur virus genome [24]. 

Approximately 1 µg of RNA was used for complementary DNA (cDNA) synthesis. In brief, cDNA was performed using Superscript IV reverse transcriptase (Thermo fischer, Waltham, MA, USA) according to the manufacturer’s guidelines. The 50 µL-PCR reaction mixture consisted of 20 pmoles each of the forward and reverse primers [23], 20 mM dNTP mixture, 1.5 mM MgCl_2_, 5X Taq DNA polymerase reaction buffer, 1.25 U of Takara Taq DNA polymerase (Takara, Nojihigash, Japan), sterile nuclease-free water and two microliters of cDNA (template) and was thermal-cycled according to the parameters as described previously [7]. The PCR products were purified using spin columns and sequenced in both directions to check for correctness of the sequencing reactions using Sanger’s dideoxy chain termination sequencing chemistry (Inqaba Biotechnologies, Pretoria, South Africa).

The edited nucleotide sequences of the generated amplicons from the highly variable glycoprotein and G-L intergenic region was undertaken on high quality amplicons, aligned in ClustalW [25] and phylogenetic trees were reconstructed using the neighbor-joining (NJ) method [26]. The topology of the reconstructed trees was validated with 1000 bootstrap replicates [27], and 70% was considered as the cut-off value supporting a phylogenetic grouping.

## 3. Results

### 3.1. Direct Fluorescent Antibody Testing and Monoclonal Antibody Typing

The honey badger FITC-stained smears demonstrated typical apple green fluorescing particles under UV-fluorescence in 100% of the smear (+4) against a reddish background (due to counter-staining) (Figure 2). The lyssavirus originating from the honey badger was shown to be a typical canid RABV with an epitope that reacts with mAb 32GD12 (Table 2).

### 3.2. RT-PCR, Sequencing and Phylogenetic Analysis

Total RNA extractions, reverse transcription PCR, nucleotide sequencing and phylogenetic reconstruction were performed and provided expected results. The topology of both the NJ and ML trees were similar (only the NJ tree shown, Figure 3). The phylogenetic analysis revealed eight separate clusters A to H with bootstrap values ranging from 73 to 100%, respectively (Figure 3). Cluster A comprised viruses obtained from Gauteng and North-West provinces of South Africa. Further, the viral isolate from a honey badger clustered with rabies viruses obtained from a domestic dog (217/21) and black-backed jackal species (Figure 3) demonstrating a single canid rabies variant easily exchanged between domestic and wildlife hosts (the branch supporting these taxa had a 100% bootstrap value). In addition, the rabies virus from the honey badger had 100% nucleotide sequence identity with viruses obtained from both domestic dog and black backed jackals, respectively. This cluster (A) was found to be distinct from lineages responsible for the rabies outbreak in Gauteng in 2011 and the most recent jackal rabies outbreak in 2016 (Figure 3). Cluster B consisted of RABV viruses obtained from domestic dogs from a previous outbreak that occurred in Gauteng province in 2011, with a bootstrap value of 94% (Figure 3). Cluster C comprised RABV viruses obtained from Lesotho, whilst clusters D and E comprised viruses obtained from the Eastern Cape province of South Africa, except for one isolate from KwaZulu Natal (KZN) province, respectively. Cluster F consists of RABV viruses from Mozambique, with the exception of one isolate from KZN province of South Africa. Cluster G consisted of the RABV viruses from Limpopo province of South Africa and the neighboring country the Republic of Zimbabwe, with a bootstrap value of 88%. Cluster G could be further subdivided into two sub-clusters G I and II, of which G I consists of viruses from Limpopo province and sub-cluster G II consists of viruses from Zimbabwe and one isolate from Mozambique (Figure 3). Cluster H comprised RABV viruses from black-backed jackals in the North-West province of South Africa (Figure 3). A mongoose rabies virus, 56/06, originating from a domestic dog and a spillover rabies virus, was used as the outgroup.

## 4. Discussion

We undertook this investigation firstly to establish the origin of the RABV recovered from the honey badger, and secondly to demonstrate the association, if any, with historical domestic and wildlife rabies cycles in this part of the country. The findings from this investigation demonstrated that the RABV originating from the honey badger was part of a cluster of RABVs originating from both domestic (dogs) and wildlife (black-backed jackals) species. This highlights the opportunistic nature and capability to switch between hosts of the canid rabies biotype. This variant was previously shown to easily cross species boundaries between domestic (dog) and wildlife hosts (jackal and bat-eared foxes), and more recently the aardwolf (Onderstepoort records). Using phylogenetic reconstruction, the canid RABV variant was shown to be distinct from the lineage responsible for the dog rabies outbreak in southwestern Gauteng in 2011 [12]. The dog rabies outbreak observed in Gauteng in 2010/2011 [12] was initiated by an imported RABV-infected dog from the rabies endemic province of KwaZulu/Natal (KZN) on the eastern seaboard of the country.

More recently, a jackal rabies outbreak that occurred on the periphery of Gauteng province in 2016 [14] was shown to be genetically similar to another RABV originating from a jackal from the North-West province of South Africa, confirming the transboundary nature of this neurotropic pathogen. Using gene sequencing of the highly divergent glycoprotein and the G-L intergenic region of lyssaviruses, it was evident that the lyssavirus identified from the honey badger clustered with a dog rabies virus and RABVs originating from black-backed jackals, suggesting the ease of transmission and exchange of rabies viruses at the human/domestic/wildlife interfaces. Furthermore, these data underscored that this specific rabies outbreak was recently introduced and distinct from the 2016 jackal rabies outbreak that was reported in this same area [14]. It is evident that the honey badger RABV infection is part of the dog–wildlife (jackal) cycle currently circulating in wildlife in black-backed jackal species in the Cradle of Humankind. In southern Africa, rabies virus infections (initiated by both canid and mongoose rabies biotypes) have previously been confirmed in wildlife, including the honey badger [28,29]. More recently in Senegal, host switching of a dog-rabies variant into a honey badger was demonstrated through genetic sequencing of the RABV isolate [30].

There is evidence of RABV host-shift events involving wildlife such as the honey badger reported in Senegal [30]. Investigations have generally been mostly epidemiological in nature, and there was less emphasis on understanding the viral molecular adaptations required to establish an infection within a new host reservoir [30,31,32,33,34]. One study [35] utilized whole genome sequence approaches to investigate the host shift that apparently occurred in Turkey at the end of the twentieth century. In this study, the authors estimated the date of the host shift into foxes to within one year and investigated viral adaptation at the sub-consensus population level. During the late 1990s, a rabies epizootic in Turkey resulted in a sustained maintenance of RABV within the fox population. Using Bayesian inferences, this independent group investigated whole genome sequences from fox and dog lyssavirus-infected brain tissues from Turkey, and demonstrated that the epizootic occurred within at least one year. Further analyses indicated that the epizootic was most likely due to a host shift from locally infected domestic dogs, rather than an introduction of a novel fox or dog RABV from outside of Turkey. The potential utility of oral rabies vaccination for the control of wildlife-associated rabies in northeastern and central South Africa has already been demonstrated, although extensive studies with wider distribution of bait are needed to assess its potential impact on rabies control in wild jackals [33,36,37,38]. Such approaches could be useful to target feral or unowned dog populations on the African continent if the zero by thirty goal is to be achieved. Infection with RABV is inevitably fatal if post-exposure prophylaxis (PEP) management is not administered in a reasonable time after exposure. In this situation, a typical category III bite exposure was handled according to standard guidelines of the World Health Organization. Hence, no human fatalities were observed due to failure of the PEP.

## 5. Conclusions

The findings from this investigation demonstrate and suggest that elimination of dog-mediated human rabies will not only rely on eliminating the disease from the source (dog), but also from wildlife populations probably using oral rabies baited vaccines, in case of host switching of rabies virus infection. Trials using chicken heads may be the preferred bait type for the oral vaccination of black-backed jackals in the Cradle of Humankind, and that consideration at a large-scale campaign should be attempted to place such baits during summer and at dusk, in order to minimize uptake by nontarget species. The potential utility of oral rabies vaccination for the control of wildlife-associated rabies in northeastern and central South Africa has already been demonstrated, although extensive studies with wider distribution of bait are needed to assess its potential impact on rabies control in wild jackals.

## Figures and Tables

**Figure 2 tropicalmed-08-00186-f002:**
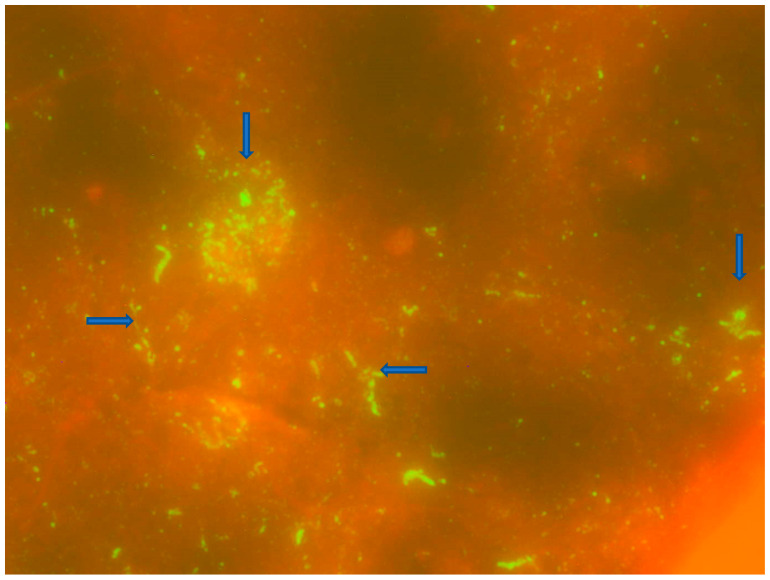
Results of the DFA staining of the honey badger brain tissues under ultra-violet (UV) fluorescence (40× magnification). The blue arrows point to the apple-green fluorescing viral inclusion bodies (Negri bodies).

**Figure 3 tropicalmed-08-00186-f003:**
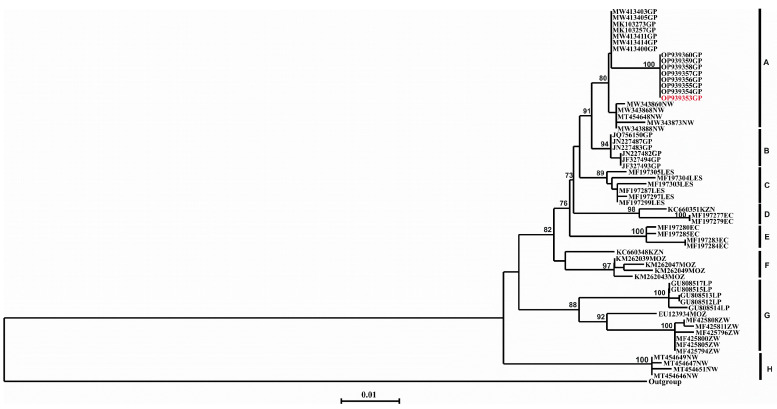
Phylogenetic analysis of RABVs analyzed in this investigation. A 592-nucleotide region encompassing the cytoplasmic domain of the glycoprotein and the G-L intergenic regions of the RABVs included in this study together with other previously characterized viruses was used in the analysis. A neighbor-joining tree of the G-L intergenic region sequences illustrating the genetic relationships of canid rabies viruses from the Kromdraai region of Gauteng. The virus sequences were compared with those from jackals from the 2016 rabies outbreak [14]. The virus sequence of the honey badger RABV is highlighted in red. The horizontal lines are proportional to the evolutionary distances between sequences and the scale bar represents nucleotide substitutions per site.

**Table 2 tropicalmed-08-00186-t002:** Reactivity patterns of the RABVs analyzed and presented here. All the RABVs exhibited a typical reactivity pattern for the canid rabies biotype or dog variant.

Mab	137/21	159/21	169/21	170/21	192/21	217/21
1C5	-	-	-	-	-	-
26AB7	+++	+++	+++	+++	+++	+++
26BE2	+++	+++	+++	+++	+++	+++
32GD12	++	+++	+++	+++	++	+++
38HF2	++	+++	+++	+++	++	+++
M612	-	-	-	-	-	-
M837	-	-	-	-	-	-
M850	-	-	-	-	-	-
M853	+++	+++	+++	++	++	+++
M1001	-	-	-	-	-	-
M1335	-	-	-	-	-	-
M1386	-	-	-	-	-	-
M1400	-	-	-	-	-	-
M1407	++	++	++	++	++	++
M1412	++	++	++	++	++	+++
M1494	-	-	-	-	-	-

-, no reactivity; ++, antigen visible in every field, but areas without infection were visible; +++, large amount of antigen visible in every microscopic field.

## Data Availability

The nucleotide sequence data generated in this study can be found on Genbank.

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
