# Peer review of "A Case Report on a Human Bite Contact with a Rabid Honey Badger Mellivora capensis (Kromdraai Area, Cradle of Humankind, South Africa)"

_tropicalmed, 2023, doi:10.3390/tropicalmed8040186_

Round 1
Reviewer 1 Report
The Paper Human bite contact with a rabid honey badger (Kromdraai area, Cradle of Humankind, South Africa) is a quite good paper reporting a case of rabies in wildlife with contact with humans.
The paper in itself is not new as a report on honey badger was published on 2022 (Faye M, Faye O, Paola ND, Ndione MHD, Diagne MM, Diagne CT, Bob NS, Fall G, Heraud JM, Sall AA, Faye O. Rabies surveillance in Senegal 2001 to 2015 uncovers first infection of a honey-badger. Transbound Emerg Dis. 2022 Sep;69(5):e1350-e1364. doi: 10.1111/tbed.14465. Epub 2022 Feb 23. PMID: 35124899.) This paper is not in the references, I advice to add it. Still is a second proof and is worth of publication.
I would advice some improvements that in my opinion would better the paper:
1. Revise the abstract according to Authors' Guidelines.
2. Introduction: paragraph from Line 54 to the end should be moved in MM in a separate section
3. Table 1 is not mentioned in the text, please add a description of the samples in the “Total RNA extractions, reverse transcription PCR (RT-PCR), nucleotide sequencing and phylogenetic reconstruction”. you used them for the tree so describe them in MM. Also in the headings you write “genetically characterized in this study” but there is no description and the study in only on one sample. If you want to keep one sample described, write that the other were sequenced for the purpose of the tree.
4. Describe the antigenic target of each Mab in Antigenic typing section or in table 2. If necessary add a figure of the virus and its antigenic regions.
5. Table 2. The tables should stand for themselves, so please explain all the Mab employed and the samples in the headings and not insert discussion or conclusion in it. Moreover there is the Mab column but not the Samples Line in the table. A line is missing between M1494 and result line.
6. Phylogenetic Tree: must be revised for two main reasons. The first is that you must use Accession numbers and not your sample ID. Second you must add other sequences deposited for your country and other countries of the continent, and also other continents, in order to give information and support your clustering. As it is it is of no use for the readers. Discussion must be improved on the new tree.
Author Response
Responses to Reviewer I
The Paper Human bite contact with a rabid honey badger (Kromdraai area, Cradle of Humankind, South Africa) is a quite good paper reporting a case of rabies in wildlife with contact with humans.
The paper in itself is not new as a report on honey badger was published on 2022 (Faye M, Faye O, Paola ND, Ndione MHD, Diagne MM, Diagne CT, Bob NS, Fall G, Heraud JM, Sall AA, Faye O. Rabies surveillance in Senegal 2001 to 2015 uncovers first infection of a honey-badger. Transbound Emerg Dis. 2022 Sep;69(5):e1350-e1364. doi: 10.1111/tbed.14465. Epub 2022 Feb 23. PMID: 35124899.) This paper is not in the references, I advice to add it. Still is a second proof and is worth of publication. Thanks for the suggestion. We have now included this reference and manuscript details in the discussion.
I would advice some improvements that in my opinion would better the paper:
- Revise the abstract according to Authors' Guidelines.
- Introduction: paragraph from Line 54 to the end should be moved in MM in a separate section: this has now been done, thank you.
- Table 1 is not mentioned in the text, please add a description of the samples in the “Total RNA extractions, reverse transcription PCR (RT-PCR), nucleotide sequencing and phylogenetic reconstruction”. you used them for the tree so describe them in MM. Also in the headings you write “genetically characterized in this study” but there is no description and the study in only on one sample. If you want to keep one sample described, write that the other were sequenced for the purpose of the tree. Thanks for your suggestion. We have now made reference to Table 1 in the test. In addition, we have also described the other viruses from black-backed jackals and domestic dogs.
- Describe the antigenic target of each Mab in Antigenic typing section or in table 2. If necessary add a figure of the virus and its antigenic regions.
- Table 2. The tables should stand for themselves, so please explain all the Mab employed and the samples in the headings and not insert discussion or conclusion in it. Moreover there is the Mab column but not the Samples Line in the table. A line is missing between M1494 and result line. This is a very good suggestion. With regards to the South African typing panel mentioned in the manuscript, it would seem that not very much work was done on trying to identify specific epitopes or even antigenic sites. With reference to Lindsay Elmgren’s thesis (1999) –the only person to work on anti-N mAb epitope mapping. Only mAb M853 was included in his studies, none of the others in the SA panel. What we know is M853 recognizes a linear epitope, and it was mapped to what he called antigenic site IV (by competitive ELISA studies), but there is no discussion/comparison to other N protein mapping studies. In brief, no epitope mapping information is available for mAbs in the panel.
- Phylogenetic Tree: must be revised for two main reasons. The first is that you must use Accession numbers and not your sample ID. Second you must add other sequences deposited for your country and other countries of the continent, and also other continents, in order to give information and support your clustering. As it is it is of no use for the readers. Discussion must be improved on the new tree. Thanks for this suggestion. The tree has now been revised.
Reviewer 2 Report
Authors aims to report the positivity of RAV in a honey badger that attacked a dog and three humans.
Specific comments/suggestion are highlighted in the attached .pdf file.
Even when the topic is of high relevance for public health and to the One Heath approach, the manuscript lack of a proper structure. It is referred to honey badger but suddenly other samples were tested. No further information is reported in terms of the exposed human population.

Author Response
Responses to Reviewer 2
- The scientific name of the honey badger has now been added.
- The result for the rabies diagnosis has now been included.
- 59 000 has now been written as 59,000.
- Thanks for this suggestion, Figure 1 - map of the are has been deleted.
- A comma has been inserted after the word HRIG,
- The word and has been inserted after anti-N Mabs,
- Reference to Table 1 has now been included in the text,
- “The Title and M&M sections only mention the honey badger, no data is reported about the other species sampled”. Although we did not include the other species in the title, we have now include them under materials and methods.
- and the meaning of the blue arrows is? Please incorporate this into the legend of figure 2. The blue arrows point to the viral inclusion bodies or Negri bodies and this has been incorporated into Figure 2.
Reviewer 3 Report
“Human bite contact with a rabid honey badger (Kromdraai area, Cradle of Humankind, South Africa)”
Thanks authors for your effort and presentation.
However, certain comments could be considered during revision:
- The title could be modified [A case report on a human bite contact with a rabid honey badger in Kromdraai area, Cradle of Humankind, South Africa)]. No follow stop should be present in the title.
- The author could include “human” in the keywords.
- Line 41, only 3 recent references are enough. The author could exclude old references.
- The sentence in lines 42-45 is so long, it should be re-phrased.
- Line 77, the reference [24] should be mentioned as the journal’s style.
- Line 80, the lesions score should be supported by a reference.
- Lines 92-94, the reactivity score for each mAb should be supported by a reference.
- Lines 132 and 166, no references should be included in the results section.
- The references underling tables and figures could be included in the Material and Methods section.
- The title of Table (2) is too long, it should be abbreviated.
- The discussion should be deeper and supported by more recent previous work.
- Line 188, what is the meaning of KZN?.
- Line 209, the conclusion section should not contain references. They could be added to the discussion or the introduction section.
- The authors should be follow the journal’s instruction for writing references.
Best wishes
Author Response
Reviewer 3:
“Human bite contact with a rabid honey badger (Kromdraai area, Cradle of Humankind, South Africa)”
Thanks authors for your effort and presentation.
However, certain comments could be considered during revision:
The title could be modified [A case report on a human bite contact with a rabid honey badger in Kromdraai area, Cradle of Humankind, South Africa)]. No follow stop should be present in the title.
Thank you for this suggestion, this has been amended.
The author could include “human” in the keywords. Thank you for this suggestion, human has now been included as a key word.
Line 41, only 3 recent references are enough. The author could exclude old references. Older references have been excluded.
The sentence in lines 42-45 is so long, it should be re-phrased. The sentence has now been rephrased.
Line 77, the reference [24] should be mentioned as the journal’s style. Thank you for this, the reference has now been aligned to the journal system of referencing.
Line 80, the lesions score should be supported by a reference. The lesions score has now been supported by a reference.
Lines 92-94, the reactivity score for each mAb should be supported by a reference.This is a fantastic suggestion. In regards to the South African typing panel, it would seem that not much work was done on trying to identify specific epitopes or even antigenic sites. Truthfully, no epitope mapping information is available for mAbs in the panel.
Lines 132 and 166, no references should be included in the results section. Thank you, the references have been removed.
The references underling tables and figures could be included in the Material and Methods section. Thanks for this, this has now been done.
The title of Table (2) is too long, it should be abbreviated. The title has been rewritten.
The discussion should be deeper and supported by more recent previous work. We have included the aspect of host switching giving two examples (from Turkey and Senegal) to demonstrate such events.
Line 188, what is the meaning of KZN?. The acronym KZN has now been written in full.
Round 2
Reviewer 1 Report
The revised paper now meet the requirements for publication in my opinion. Just one request: please declare which sequence you used as outgroup.
Author Response
A mongoose rabies virus, 56/06, originating from a domestic dog and a spillover rabies virus, was used as the outgroup.
Reviewer 2 Report
Authors aims to report the positivity of RAV in a honey badger that attacked a dog and three humans.
Specific comments/suggestion are highlighted in the attached .pdf file.
Authors satisfactorily responded to the requirements made by this reviewer

Author Response
- The scientific name of the honey badger (Mellivora capensis) was added in the title and italicised.
- section 3.2 has now been changed to vertical